# Prognostic Role of Circulating miR-141 in Early Diagnosis of Colorectal Cancer

**DOI:** 10.3390/medicina61061007

**Published:** 2025-05-28

**Authors:** Merve Simsek Dilli, Ertugrul Bayram, Ismail Oguz Kara

**Affiliations:** 1Department of Endocrinology, Adana City Training and Research Hospital, Adana 01230, Turkey; msimsekdilli@gmail.com; 2Department of Medical Oncology, Faculty of Medicine, Cukurova University, Adana 01250, Turkey; iokara@cu.edu.tr

**Keywords:** colorectal cancer, miRNA-141, cancer biomarkers, non-coding RNA, gene expression profiling

## Abstract

*Background*: Colorectal cancer (CRC) is the second leading cause of cancer-related deaths worldwide, emphasizing the urgent need for reliable biomarkers for early diagnosis and prognosis. MicroRNA-141 (miR-141) is a small non-coding RNA molecule that plays a regulatory role in cell proliferation and differentiation and has been linked to various types of cancer. *Objective*: This study aims to evaluate the clinical and prognostic relevance of circulating levels of miR-141 in untreated patients with CRC. *Method*: A total of 40 patients with CRC and 20 healthy subjects matched for age and sex were selected for this study. Blood samples from these individuals were analyzed using real-time PCR to determine the levels of miR-141. *Results*: Despite the absence of any substantial correlations between levels of miR-141 and conventional prognostic factors, including disease stage, lymph node involvement, vascular/perineural invasion, or metastasis, a statistically significant correlation was identified between miR-141 levels and the extent of local tumor invasion (T stage) (*p* = 0.034). These findings suggest that microRNA-141 may be involved in local tumor infiltration and warrant further validation in larger, multicenter studies. *Conclusions*: Although miR-141 alone may not serve as a definitive biomarker for CRC diagnosis or prognosis, its potential role—particularly in combination with other markers—could contribute to refined risk stratification strategies in CRC management.

## 1. Introduction

CRC is a major cause of morbidity and mortality worldwide and is one of the most common malignancies of the gastrointestinal tract. The incidence of the disease is higher in developed countries and accounts for approximately 10% of all cancers in men and women [1,2]. Various factors such as age, genetic predisposition, diet, and environmental factors play a role in the development of CRC. It has been reported that changes in the gut microbiota may also play a role in the development of inflammation-related malignancies, such as colorectal cancer. The inflammatory microenvironment resulting from altered microbiota composition has the potential to affect epigenetic mechanisms, particularly levels of microRNA expression [3]. In particular, genetic changes and epigenetic regulations have been the focus of interest in understanding the molecular mechanisms of the disease [2,4].

In recent years, the role of microRNAs (miRNAs) in CRC development and progression has received increasing attention. MicroRNAs are small, non-coding RNA molecules that play critical roles in the regulation of cellular processes and control gene expression at the post-transcriptional level [5,6]. The first detection of miRNA expression changes in Chronic Lymphocytic Leukemia (CLL) revealed the importance of these molecules in cancer biology. Subsequent studies have shown that miRNAs may play oncogenic or tumor suppressor roles in various cancer types [6,7].

In the context of colorectal cancer, certain microRNAs (e.g., miR-21, miR-31, and miR-92a) are overexpressed, while others (e.g., miR-143, miR-145, and the let-7 family) are found at low levels [8,9]. These differences play a critical role in the regulation of cellular pathways involved in tumor formation, progression, and metastasis. Specifically, the microRNA miR-21 has been demonstrated to enhance tumor invasion and metastasis, and its expression has been correlated with unfavorable prognoses by repressing programmed cell death-4 (PDCD4) in colorectal cancer patients [10,11].

microRNA-141 (miR-141), located on chromosome 12, has been shown to play a prominent role, particularly in cases of epithelial cell type tumors. Studies have indicated that abnormal expression of miR-141 may have an impact on tumor progression, metastasis, and the clinical course of the disease [12,13]. miR-141 expression is downregulated in gastric cancer tissues, which may be related to its inhibitory effect on cell proliferation [14]. In contrast, elevated levels of miR-141 have been found in ovarian, prostate, and gallbladder cancers and have been correlated with poor prognosis [15,16]. In the context of CRC, studies have indicated that miR-141 may contribute to the processes of invasion and metastasis, thereby offering significant insights into the progression of the disease [14,17]. Cheng et al. demonstrated that elevated levels of miR-141 in plasma samples from CRC patients were associated with diminished survival rates, suggesting that miR-141 may serve as a prognostic factor [18].

This study uniquely focuses on evaluating the clinical relevance of miR-141 in untreated CRC patients, addressing a gap in the literature by examining its role in early diagnosis and as a potential prognostic biomarker. By incorporating a comprehensive analysis of miR-141 expression profiles, this study offers novel insights into the development of personalized treatment strategies, aiming to improve CRC management and patient outcomes.

## 2. Materials and Methods

### 2.1. Study Design and Participants

The present study was conducted on patients admitted to the Çukurova University Faculty of Medicine and who had received a pathological diagnosis of colorectal cancer. Written informed consent was obtained from patients who consented to participate in this study. This study was approved by Çukurova University Clinical Research Ethics Committee.

A comprehensive set of demographic and clinical data were collected, encompassing various demographic characteristics such as age, gender, and lifestyle habits, including smoking and alcohol use. Additionally, detailed cancer-related information was documented, including cancer stage, lymph node involvement, vascular invasion, perineural invasion, metastasis status, family history of cancer, and carcinoembryonic antigen (CEA) level. The control group comprised age- and gender-matched healthy individuals. The study sample included a total of 40 patients and 20 control subjects, with a 2:1 match ratio.

Our primary objective was to evaluate miR-141 levels at the time of diagnosis, prior to any therapeutic intervention, to assess its potential role as a diagnostic and early prognostic biomarker.

### 2.2. Inclusion and Exclusion Criteria

This study included patients diagnosed with CRC who had not received chemotherapy or radiotherapy prior to blood sampling. The inclusion criteria were as follows: (1) histopathologically confirmed CRC diagnosis, (2) availability of sufficient peripheral blood samples for molecular analysis, and (3) no history of other malignancies or chronic inflammatory diseases (Figure 1).

The control group consisted of age- and sex-matched healthy individuals with no history of malignancy, chronic inflammatory disease, or gastrointestinal disorders.

The exclusion criteria were as follows: (1) previous or ongoing chemotherapy or radiotherapy, (2) presence of any other malignancy or systemic inflammatory disease, (3) insufficient blood sample quality or volume for molecular analysis, and (4) refusal to provide informed consent.

### 2.3. RNA Extraction

A total of 2.5 milliliters of peripheral blood was collected from each participant and placed in tubes containing RNA Later solution. The tubes were stored at −20 °C until the day of analysis. RNA isolation was performed using a High Pure miRNA isolation kit according to the manufacturer’s protocol. During the isolation process, blood samples stored in the RNA Later tube were thawed at room temperature for one hour. Subsequently, the leukocyte count was determined, and the blood sample was separated into 3,000,000 cells. The samples were treated twice with red blood cell lysis buffer to remove erythrocytes and then subjected to centrifugation to obtain a leukocyte pellet. Cell lysis was achieved by adding lysis binding buffer to the leukocyte pellet. For RNA isolation, washing and elution steps were performed using filter tubes. The isolated total RNA was stored at −20 °C.

RNA quantification was indeed performed using a NanoDrop spectrophotometer to ensure that the extracted RNA from each sample met the quality and concentration requirements for downstream real-time PCR analysis. Only samples with sufficient RNA yield and acceptable purity (A260/280 ratio between 1.8 and 2.1) were included in the analysis.

The synthesis of cDNA was carried out using the miRCURY Universal cDNA synthesis kit. The reaction mixture was prepared by adding 4 μL of 5× reaction buffer, 9 μL of nuclease-free water, 2 μL of enzyme mixture, 1 μL of synthetic spike-in control, and 4 μL of total RNA (5 ng/μL) to the appropriate tubes. The cDNA synthesis reaction, with a total volume of 20 μL, was carried out within a 96-well plate of a Light Cycler 480 (Roche, Basel, Switzerland) system. The reaction was subjected to a protocol consisting of 60 min at 42 °C, 5 min at 95 °C, and storage at +4 °C or long-term storage at −20 °C.

### 2.4. Real-Time PCR

Real-time PCR analysis was performed with SYBR Green Master Mix using a Light Cycler 480 system. For normalization of miRNA expression levels, SNORD48 was used as the endogenous reference control and hsa-miR-141 was used as the target gene. The PCR mix for the reference gene was prepared as follows: 5 μL SYBR Green Master Mix, 1 μL SNORD48 PCR Primer Mix, and 4 μL 1/80 diluted cDNA in a total volume of 10 μL. The PCR mix for the target gene was prepared as follows: 5 μL SYBR Green Master Mix, 0.5 μL hsa-miR-141 PCR Primer Mix (Forward), 0.5 μL hsa-miR-141 PCR Primer Mix (Reverse), and 4 μL 1/80 diluted cDNA in a total volume of 10 μL.

The amplification process was carried out using a real-time quantitative polymerase chain reaction (qPCR) system based on SYBR Green detection chemistry, which allows the quantification of DNA amplification in real time through fluorescence monitoring. The amplification was performed on the LightCycler 480 system (Roche, Basel, Switzerland), which uses a high-resolution optical system to detect SYBR Green fluorescence with high sensitivity and reproducibility. During each cycle, the SYBR Green dye intercalates into the double-stranded DNA products, and the increase in fluorescence signal is measured at the end of each extension step. This allows for the continuous monitoring of product accumulation. To ensure specificity of the amplification, melting curve analysis was performed immediately after the completion of PCR cycles. The melting curve protocol included a denaturation step at 95 °C for 30 s, a rapid cool-down to 40 °C for 1 min to allow reannealing of DNA strands, followed by a gradual increase to 85 °C during which fluorescence was continuously measured. The presence of a single sharp peak in the melting curve confirmed the specificity of each amplified product, while the absence of primer-dimer artifacts or non-specific amplification was verified by the curve profiles.

All the reactions were performed in triplicate to ensure reproducibility and accuracy. Amplification efficiency and cycle threshold values were analyzed using the instrument’s software, and relative expression levels of hsa-miR-141 were calculated using the ΔΔCt method with SNORD48 as the endogenous control.

### 2.5. Statistical Analysis

All the statistical analyses were performed using IBM SPSS 19. The demographic and clinical data of the patient and control groups were summarized using descriptive statistics. The distribution of CEA and miR-141 levels in different groups was analyzed using the Mann–Whitney U test. The Kruskal–Wallis test was used to evaluate the relationship with T, N, M stages, and post hoc analysis was performed using the Bonferroni-corrected Mann–Whitney test when significant differences were found. The diagnostic accuracy of CEA and miR-141 was determined by ROC analysis, which involved generating ROC curves and calculating the area under the curve (AUC). The sensitivity and specificity ratios were calculated with the threshold values, and the potential biomarker role of miR-141 in the diagnosis of colorectal cancer was evaluated.

## 3. Results

### Patient Characteristics

This study encompassed 40 patients diagnosed with colorectal cancer and 20 healthy volunteers who were matched on the basis of age, gender, and smoking status. Within the patient group, the distribution was as follows: 7 patients were classified as stage I, 9 as stage II, 12 as stage III, and 12 as stage IV.

In the context of age-based group comparisons, the mean age of the patients was determined to be 61.15 years ± 14.1, while that of the control group was found to be 59.1 years ± 13.3 (Table 1). Upon direct comparison of these two groups with respect to age, no statistically significant difference was observed (*p* = 0.592).

A comparison of the two groups reveals that there was a 50% representation of female patients, while the male demographic constituted 50% of the total. Within the designated control group, the numerical distribution of female patients was 9, constituting 45% of the total, while the male demographic accounted for 11 patients, amounting to 65%.

Table 2 presents the distribution of colorectal cancer patients according to tumor localization. Colon cancer was detected in 65% of patients, whereas rectal cancer was detected in 35%.

The median value of the miR-141 level in the patient group was determined to be 0.00016, while the median value of the miR-141 level in the control group was 0.000057. The mean CEA level of the patients and the control group was 11.2 ± 19.4 and 1.8 ± 0.88, respectively (Table 3). There was no statistically significant difference between the patient and control groups in terms of CEA levels (*p* = 0.132).

A significant correlation was identified between disease stage and CEA level (Figure 2) when the two variables were compared (*p* = 0.009).

No significant difference was found between miR-141 levels when the patient and control groups were compared (*p* = 0.612).

A comparison of miR-141 levels according to disease stage (Figure 3) revealed no statistically significant differences (*p* = 0.219).

CEA and miR-141 levels were compared with prognostic factors of colorectal cancer patients. Vascular invasion was observed in 75% of the patients. A comparison of vascular invasion, a prognostic factor for colorectal cancer, with the level of miR-141 revealed no significant difference (*p* = 0.292). No significant difference was found when vascular invasion was compared with CEA level (*p* = 0.087).

Perineural invasion was observed in 17.5% of colorectal cancer patients. A comparison of perineural invasion with miR-141 levels revealed no significant difference (*p* = 0.681). Similarly, no significant difference was identified when perineural invasion was compared with CEA levels (*p* = 0.650).

When classified according to the local tumor invasion level (T), 20% of the patients were T1 and T2, 30% were T3, and 50% were T4. A comparison was made between CEA and miR-141 levels according to the local tumor invasion level (T). A significant correlation was identified between local tumor invasion level and CEA level (Figure 4), contingent upon the local tumor invasion level (*p* = 0.039). The miR-141 level was compared according to the local tumor invasion level (Figure 5). A significant correlation was found between local tumor invasion level and miR-141 (*p* = 0.034).

According to the TNM staging system, lymph node involvement was classified as N0 in 42.5% of cases, N1 in 50%, and N2 in 7.5%. A comparison was made between CEA levels and miR-141 levels according to lymph node involvement (N). For the purpose of this analysis, N1 and N2 lymph node involvement were combined. A significant correlation was found between CEA level and lymph node involvement (Figure 6), (*p* = 0.017). Conversely, no significant correlation was identified between the presence of lymph node involvement and the level of miR-141 (*p* = 0.221).

This study also noted that 30% of patients exhibited metastasis. Subsequently, CEA and miR-141 levels were compared according to metastasis status. A significant correlation was identified between CEA level and the presence of metastasis (Figure 7), (*p* = 0.017). Conversely, when the presence of metastasis was compared with the level of miR-141, no significant correlation was identified between the presence of metastasis and the level of miR-141 (*p* = 0.149).

The correlation coefficient of the relationship between CEA and miR-141 was obtained, yielding a value of R equal to −0.05 (*p* = 0.71) (Figure 8).

The area under the curve was calculated for miR-141, yielding an AUC = 0.54 (*p* = 0.65) as a result of ROC Curve Analysis. When 0.000029 was designated as the threshold value for miR-141, the sensitivity and selectivity of miR-141 above 0.000029 were found to be 40% and 75%, respectively (Figure 9A).

The CEA was calculated as AUC = 0.62 (*p* = 0.13) as a result of ROC Curve Analysis, and, when 3 was taken as the threshold value for CEA, sensitivity and selectivity were found to be 40% and 85%, respectively, for CEA above 3 (Figure 9B). Table 4 comprehensively summarizes the key findings of the study.

## 4. Discussion

Colorectal adenocarcinoma represents the most prevalent form of cancer within the gastrointestinal tract. It is a significant cause of morbidity and mortality on a global scale. Environmental and genetic factors play a pivotal role in the development of colorectal cancer [2,19].

MicroRNAs are transcribed from highly conserved DNA regions but do not encode proteins. These non-protein-coding RNA molecules bind to target mRNAs that are complementary to their nucleotide sequence, thereby regulating post-transcriptional gene expression through mechanisms such as translational repression or mRNA degradation [20]. MiRNAs have been identified in various cancer types, including colorectal cancer. These molecular units have emerged as promising biomarkers, offering novel insights into tumor characterization and facilitating predictions regarding responses to different active chemotherapies [21].

miRNAs have demonstrated considerable promise as diagnostic and therapeutic instruments in the context of digestive system diseases. The regulatory functions of these elements in the context of inflammation, fibrosis, tumor progression, and metabolism have prompted intensive investigation [22].

Studies have reported that miRNA expression can be used as a diagnostic marker in colorectal cancer [23]. In our study, we aimed to determine the clinical prognostic significance of miRNA-141 level in colon cancer, to use miRNA-141 level as a tumor marker in early diagnosis of colon cancer, to investigate its role in early diagnosis and recurrence, and to evaluate its correlation with CEA. The aim was to demonstrate high miRNA-141 levels in metastatic colorectal cancer, to correlate miRNA-141 levels with the prognosis of the disease, and to evaluate it as an indicator of early recurrence and tumor burden.

Chong et al. reported that miR-92 levels were significantly elevated in colorectal cancer patients and could be used as a non-invasive marker for the diagnosis of colorectal cancer [24]. In a study by Nishida et al. in 89 colorectal cancer patients, high miR-125b levels were associated with tumor size, tumor invasion, and poor prognosis [25].

Wang et al. showed that low levels of miR-195 were associated with lymph node metastasis and advanced tumor stage [26]. Huang and Pu showed that miR-29a and hsa-miR-221 were significantly overexpressed in all stages of colorectal cancer patients compared to the healthy group [27]. The findings from these studies corroborate the hypothesis that the use of miRNA profiling as a diagnostic biomarker is merited further validation.

Our primary objective was to evaluate miR-141 levels at the time of diagnosis, prior to any therapeutic intervention, to assess its potential role as a diagnostic and early prognostic biomarker. However, many previous studies did not evaluate the diagnostic and prognostic role of miRNAs in conjunction with classical biomarkers such as CEA and often lacked comprehensive analysis across tumor stages and invasion parameters. Moreover, the sample sizes in prior reports were frequently limited, and standardized follow-up strategies were often absent, making comparisons difficult.

The utilization of CEA is not recommended for the diagnosis or screening of colorectal cancer, owing to its high rate of false positives. Its primary applications are in the evaluation of response to treatment and the detection of recurrences in colon cancer. It is recommended that CEA levels be measured at the commencement of treatment and subsequently every 2–3 months during active treatment [28].

Although no statistically significant difference was observed (*p* = 0.132), the trends observed in miR-141 and CEA levels suggest a potential association that warrants further investigation in larger cohorts.

It is widely acknowledged that tumor stage constitutes the most significant prognostic factor in colon cancer [29]. Cheng et al. found elevated levels of miR-141 in patients with stage IV colon cancer. This study demonstrated that elevated levels of plasma samples from 156 patients were associated with diminished survival outcomes and could serve as a prognostic factor in the development of colon cancer [17].

In our study, the levels of miR-141 and CEA were examined in relation to colorectal cancer prognostic factors, including vascular invasion, stage, perineural invasion, metastasis, lymph node involvement, and local tumor invasion. miR-141 and CEA levels showed a stage-independent profile in our cohort, indicating that their expression might be influenced by additional biological or molecular factors beyond traditional staging.

Perineural invasion, a hallmark of advanced colorectal cancer, has been identified as a significant prognostic factor [2]. No statistically significant discrepancy was observed when comparing the levels of miR-141 and CEA in patients with and without perineural invasion in this study.

In the presence of vascular invasion, the five-year survival rate is significantly reduced. Consequently, vascular invasion is regarded as a prognostic factor [30]. While some parameters did not show statistically significant correlations, consistent patterns and directionality of the findings point to a biological relevance that could become evident in studies with greater statistical power.

In the context of colon cancer, the presence of lymph node metastasis constitutes a pivotal prognostic factor [31]. In the present study, a comparative analysis was conducted between the levels of miR-141 and CEA with respect to the presence of lymph node metastasis. The findings revealed an absence of a statistically significant correlation between miR-141 levels and lymph node metastasis. However, a significant correlation was identified between CEA levels and the presence of lymph node metastasis (*p* = 0.017).

Local tumor spread has been demonstrated to be associated with prognosis in colorectal cancers. A worse prognosis is expected in cases of high T stage [32]. In the present study, a significant correlation was identified between local tumor invasion and levels of both miR-141 (*p* = 0.034) and CEA. In addition, a significant correlation was identified between CEA levels and local tumor invasion (*p* = 0.017).

Numerous studies have demonstrated the critical role that miR-141 plays in cancerous cell migration, invasion [33], and cisplatin resistance [34]. A comparison was made between the levels of miR-141 and CEA with the presence of metastasis. No significant correlation was identified with miR-141, while a significant correlation was found between CEA level and the presence of metastasis (*p* = 0.017). Furthermore, a substantial correlation was identified between CEA levels and various prognostic factors, including metastasis, lymph node involvement, and tumor invasion level. This finding underscores the critical role of CEA in prognostic assessment.

In the ROC analysis, miR-141 had an AUC value of 0.54 (*p* = 0.65), showing low sensitivity (40%) and moderate selectivity (75%). The AUC for CEA was 0.62 (*p* = 0.13), with a sensitivity of 40% and selectivity of 85% when a cut-off value of 3 ng/mL was used. Despite its relatively low sensitivity, the high specificity of CEA remains clinically valuable, and its utility could be enhanced through complementary biomarkers such as miR-141. There was no significant correlation between miR-141 and CEA (*p* = 0.71), suggesting that they are biologically dependent on different mechanisms.

The observed variations among the series can be attributed to several factors, including heterogeneity in tumor stage, localization, genetic background, and technical nuances. While miR-141 on its own may not fully capture the complexity of colorectal cancer prognosis, its integration into multi-marker panels holds promise for more comprehensive prognostic models. In the present study, a significant correlation was identified between the level of miRNA-141 and T, which is a prognostic factor in colon cancer (*p* = 0.034). No significant correlation was observed with other prognostic factors.

### Strengths and Limitations

The strengths of this study lie in its focus on miR-141’s specific role in CRC, its novel approach in evaluating circulating miR-141 levels, and its potential for contributing to more targeted therapeutic interventions.

The relatively small sample size, particularly in the control group, may limit the statistical power and generalizability of the findings. Given the heterogeneity of treatment timelines and clinical stages, a standardized follow-up for outcome prediction was not feasible within the current study design. These factors collectively suggest that, while the findings are promising, they should be interpreted cautiously, and further multicenter studies with larger cohorts and mechanistic evaluations are warranted to validate and expand upon our results.

While miR-141 alone may not provide sufficient sensitivity for use as a standalone biomarker, its potential value as a component of a multi-marker panel in colorectal cancer prognosis and recurrence monitoring is promising. Our findings highlight the need for further large-scale, prospective, and mechanistic studies to validate these observations and explore miR-141’s functional role in colorectal tumorigenesis.

## 5. Conclusions

miR-141 may have limited value alone but could enhance prognostic accuracy when combined with CEA in patients with CRC. Further large-scale studies are needed to confirm its clinical potential.

## Figures and Tables

**Figure 1 medicina-61-01007-f001:**
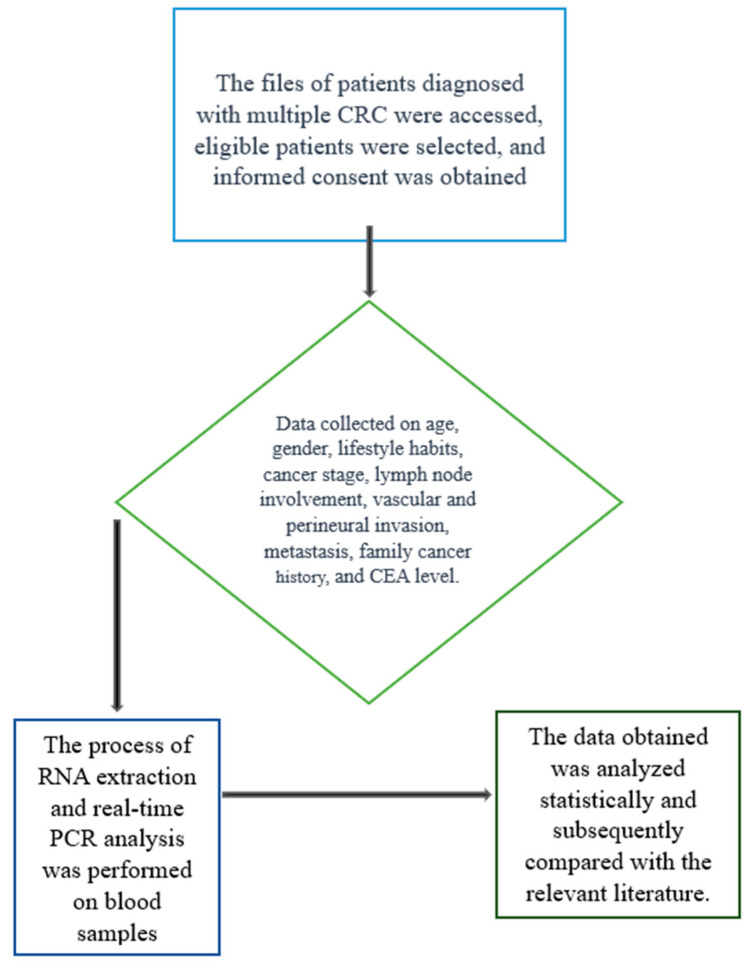
Flowchart to illustrate the study design.

**Figure 2 medicina-61-01007-f002:**
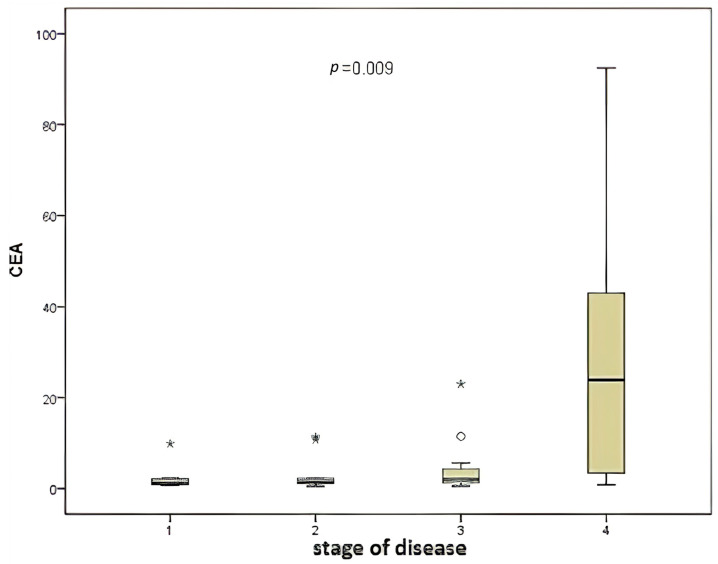
Comparison of disease stage and CEA level.

**Figure 3 medicina-61-01007-f003:**
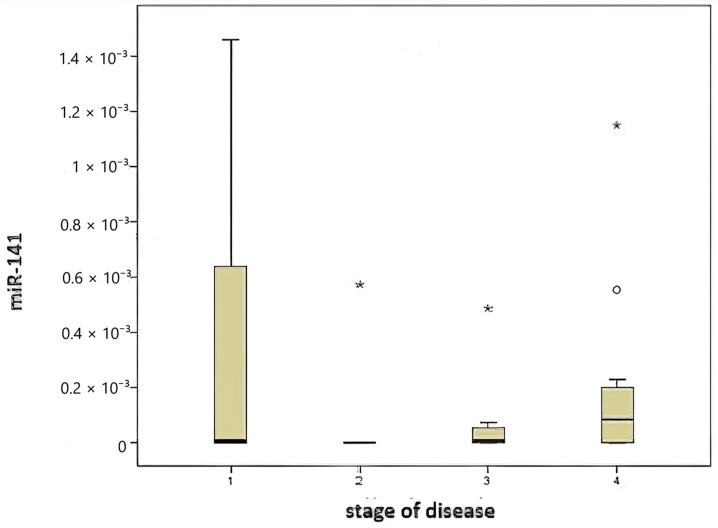
Comparison of disease stage and miR-141 level.

**Figure 4 medicina-61-01007-f004:**
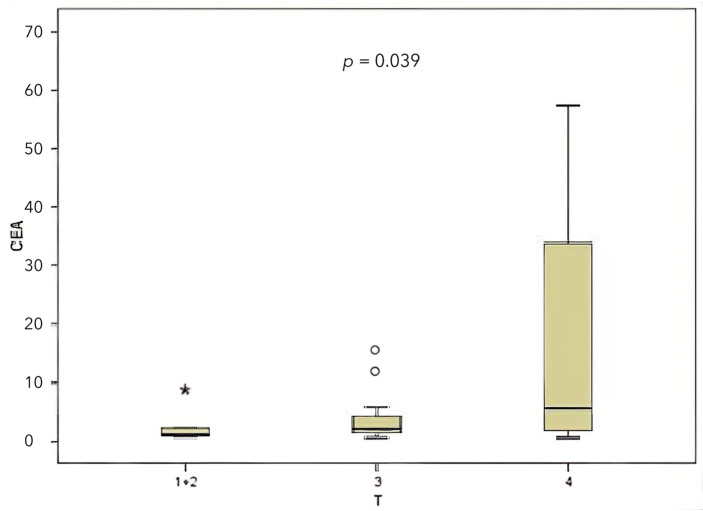
Comparison between the local tumor invasion level and the CEA level.

**Figure 5 medicina-61-01007-f005:**
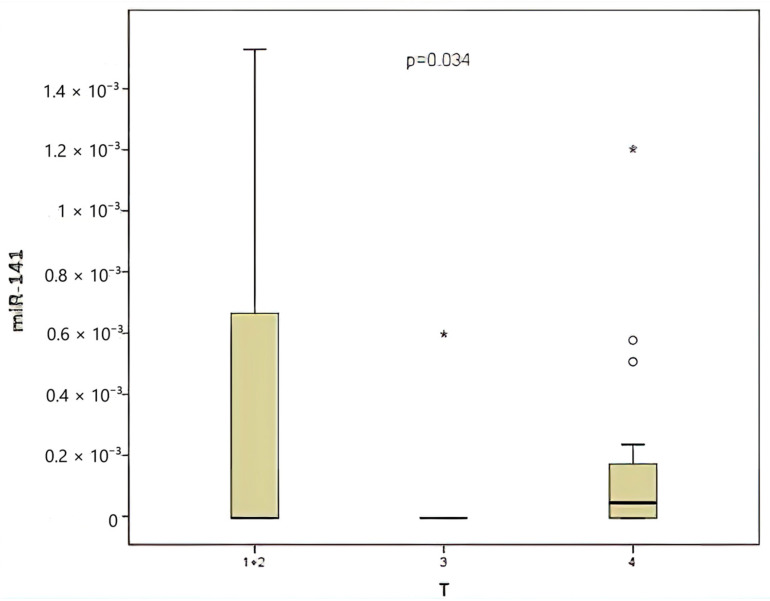
Comparison between the local tumor invasion level and the miR-141 level.

**Figure 6 medicina-61-01007-f006:**
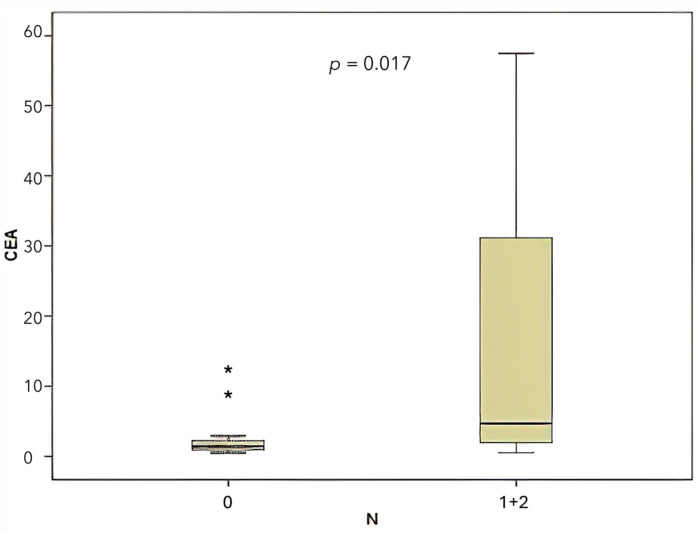
Comparison of CEA levels according to lymph node involvement (N).

**Figure 7 medicina-61-01007-f007:**
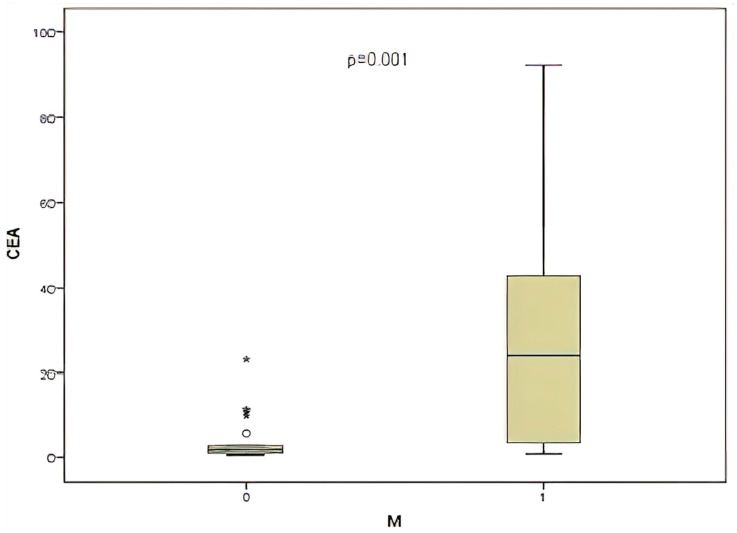
Comparison of CEA level and presence of metastasis (M).

**Figure 8 medicina-61-01007-f008:**
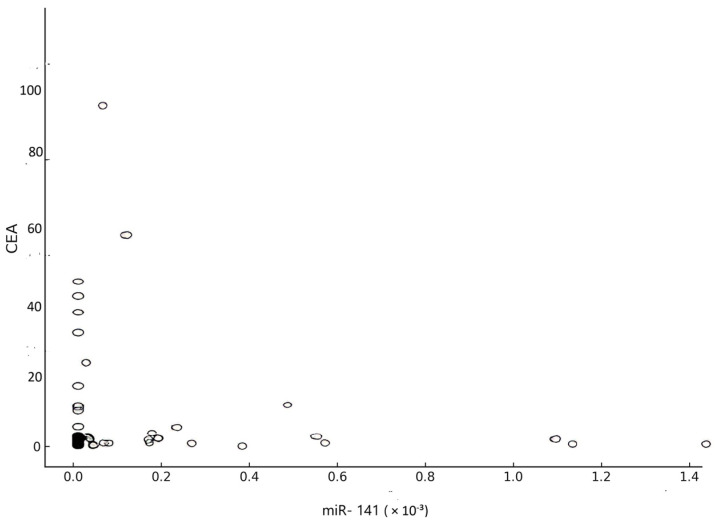
Correlation between CEA and miR-141.

**Figure 9 medicina-61-01007-f009:**
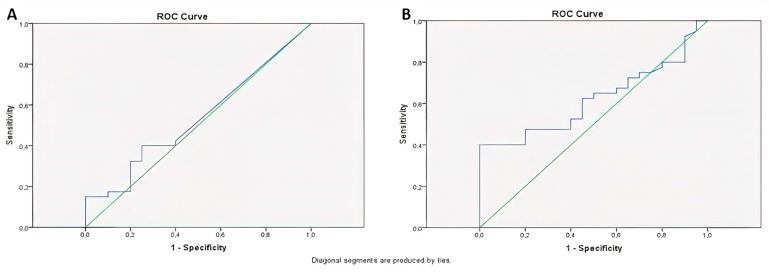
The analysis of the ROC curve for miR-141 (**A**) and CEA (**B**).

**Table 1 medicina-61-01007-t001:** Demographic and clinical characteristics of the patient and control group.

Variable	Category	Patients (*n* = 40)	Controls (*n* = 20)	Total (*n* = 60)
Gender	Female	20 (50.0%)	9 (45.0%)	29 (48.3%)
	Male	20 (50.0%)	11 (55.0%)	31 (51.7%)
Age (years)	Mean ± SD	61.15 ± 14.1	59.10 ± 13.3	60.47 ± 13.8
Tumor Stage	Stage I	7 (17.5%)	-	7 (11.7%)
	Stage II	9 (22.5%)	-	9 (15.0%)
	Stage III	12 (30.0%)	-	12 (20.0%)
	Stage IV	12 (30.0%)	-	12 (20.0%)

**Table 2 medicina-61-01007-t002:** Patients according to tumor localization.

		Number	%	Percentage of Patients	Cumulative Percentage
Patient	Colon	26	43.3	65.0	65.0
	Rectum	14	23.3	35.0	100.0
	Total	40	66.7	100.0	
Control		20	33.3		
Total		60	100		

**Table 3 medicina-61-01007-t003:** The levels of CEA and miR-141 in patients and the control group.

	miR-141 Median (Min–Max)	CEA Median (Min–Max)	*p*
Patient	0.00016 (0.000000–0.001460)	11.23 (0.49–92.47)	
Control	0.000057 (0.000000–0.000382)	1.85 (0.30–3.30)	0.612

**Table 4 medicina-61-01007-t004:** Summary of key findings.

Variable	Comparison	Result	*p* Value	Interpretation
CEA	Patients vs. Controls	11.2 ± 19.4 vs. 1.8 ± 0.88	0.132	Not significant
CEA	By Disease Stage	Higher CEA in advanced stages	0.009	Significant
miR-141	Patients vs. Controls	0.00016 vs. 0.000057	0.612	Not significant
miR-141	By Disease Stage	No significant difference	0.219	Not significant
CEA	With Vascular Invasion	Higher in invasive cases	0.087	Not significant
miR-141	With Vascular Invasion	No significant difference	0.292	Not significant
CEA	With Perineural Invasion	No significant difference	0.650	Not significant
miR-141	With Perineural Invasion	No significant difference	0.681	Not significant
CEA	T	Higher CEA in T4	0.039	Significant
miR-141	T	Higher in T4	0.034	Significant
CEA	N	Higher with N1/N2	0.017	Significant
miR-141	N	No significant difference	0.221	Not significant
CEA	With Metastasis	Higher in metastatic cases	0.017	Significant
miR-141	With Metastasis	No significant difference	0.149	Not significant
CEA vs. miR-141	Correlation	R = −0.05	0.71	Not significant
ROC Analysis	miR-141	AUC = 0.54	0.65	Low diagnostic value
ROC Analysis	CEA	AUC = 0.62	0.13	Moderate diagnostic value

## Data Availability

The data are available on request from the authors.

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
