# Peer review of "Prognostic Role of Circulating miR-141 in Early Diagnosis of Colorectal Cancer"

_medicina, 2025, doi:10.3390/medicina61061007_

Round 1

Reviewer 1 Report

Comments and Suggestions for Authors

In my opinion the Ms is relevant and deserve to be published. However, some points need to be revised before the manuscript can be published.

- The title needs to be clearly defined/formulated.

- The abstract should be revised to better highlight the significance of the topic.

- Include keywords that are not already present in the title.

- Please add a list of abbreviations.

- Introduction: if a word has an abbreviation and already cited one time, it is enough to use this abbreviation in the all text.

- Introduction: it is necessary to improve the introduction with recent references, also add more information about the studied disease.

- The authors should justify the selected sample number (40 CRC patients and 40 controls). Is this number sufficient to draw robust conclusions about the prognostic value of miR-141?

- Authors should specify the selection criteria and the method for matching healthy controls to patients. Were factors such as inflammatory status or the presence of other pathologies taken into account?

- The control group is stated to be 20 individuals to 40 patients, a ratio of 2:1. However, the abstract states a 1:1 matching (40 patients and 40 controls). Could you please verify?

- Patients recruited from a single hospital may not be representative of the general colorectal cancer patient population, which could introduce selection bias. Please check

- The groups may not be balanced in terms of gender, age, and lifestyle factors, and it would be important to statistically check for homogeneity between the groups.

- It is important to perform RNA quantification tests to ensure that the extracted amount is sufficient for downstream analyses.

- Storing isolated RNA at -20°C can lead to degradation over time. What is the maximum storage time allowed for the RNA before analysis, and have tests been performed to verify the integrity of the RNA after extended storage?

- All p in the manuscript should be in italics and capital throughout the manuscript.

- More detailed discussion of the study's limitations would help readers understand the scope and application of the findings.

- The conclusions regarding the improvement in prognostic accuracy with miR-141 and CEA are based on preliminary data. Are complementary analyses or multicenter studies planned to validate these results before recommending the clinical use of this biomarker combination?

- The conclusion should be summarized and rewritten.

- The figures are completely blurred, please redraw them.

- Add the statistical tests to the table.

Author Response

Dear Reviewer,

We would  like to thank you for the suggestions and comments. We made changes that were detailed below. All revisions and improvements are highlighted yellow in the revised version of our manuscript. We would like to thank you again for your valuable time and comments for strengthen our paper.

Best regards,

We have revised title to better reflect the scientific content of the study.

The abstract has been revised to better emphasize the significance and potential implications of the study findings.

Additional keywords not present in the title have been included as requested.

A list of abbreviations has been added.

The manuscript has been revised accordingly—abbreviations are now used consistently throughout the text after their first mention.

We have revised the Introduction section by including more detailed information about colorectal cancer and its clinical significance. Additionally, we have updated the references and incorporated recent literature to strengthen the context and relevance of the study. We hope these improvements address your comments appropriately.

The sample size of 40 colorectal cancer patients and 20 age- and gender-matched healthy, non-smoking controls was determined based on logistical and financial limitations, as well as the availability of eligible participants. While we acknowledge that the current sample size may limit the generalizability of the findings, we have also included a post hoc power analysis to evaluate the statistical power of our findings, we believe the findings offer a foundation for future large-scale studies.

The control group consisted of healthy individuals who were age- and sex-matched with the colorectal cancer patients. All controls were non-smokers and had no known history of chronic diseases, malignancies, or acute infections at the time of sample collection. Individuals with any signs of systemic inflammation or other pathological conditions were excluded based on medical history and basic clinical evaluation. These criteria were implemented to minimize potential confounding factors and to ensure the comparability of the groups.

We apologize for the confusion. The correct number of participants in the control group is 20, not 40, resulting in a 2:1 ratio with the 40 colorectal cancer patients. This has been corrected in the abstract and related sections of the manuscript.

We acknowledge that recruiting patients from a single center may pose a risk of selection bias. However, the patients included in our study were recruited from a large regional university hospital that serves a broad and diverse population. As a tertiary care center, it receives referrals from a wide geographic area and includes patients with various demographic and clinical characteristics. Therefore, we believe the study population provides a reasonably representative sample of colorectal cancer patients in our region.

In our study, control subjects were age- and sex-matched as closely as possible to the patient group. Additionally, we performed statistical analysis to assess homogeneity between the two groups in terms of age and gender, and no significant differences were found.

In our study, RNA quantification was indeed performed using a NanoDrop spectrophotometer to ensure that the extracted RNA from each sample met the quality and concentration requirements for downstream real-time PCR analysis. Only samples with sufficient RNA yield and acceptable purity (A260/280 ratio between 1.8 and 2.1) were included in the analysis.

In our study, all RNA samples were stored at -20°C for a maximum of two weeks before analysis. To minimize degradation risk, RNA was extracted and stored in RNA later solution immediately after blood collection. Prior to downstream applications, RNA purity and concentration were evaluated using a NanoDrop spectrophotometer.

We have carefully revised the manuscript to ensure that all p-values are now written in italics and capital.

This study has several limitations that should be acknowledged. These limitations underscore the need for larger, multicenter, prospective studies with longitudinal follow-up to validate the current findings and to better assess the prognostic value of miR-141 in colorectal cancer.

We aim to collaborate with other institutions to conduct multicenter studies. These efforts will help strengthen the statistical power, improve the generalizability of the findings, and provide more robust evidence regarding the clinical applicability of this biomarker combination.

We have revised the conclusion section as suggested to make it more concise and focused.

The figures have been redrawn with improved resolution and clarity, and the updated versions have been included in the revised manuscript.

We have added the appropriate statistical test results to the relevant tables and clarified them in the revised manuscript.

Reviewer 2 Report

Comments and Suggestions for Authors

This is an interesting study exploring emerging biomarkers in the early diagnosis of colorectal cancers, as the second most prevalent cancer worldwide. The data provided are valuable and could have a significant impact on future colorectal cancer diagnosis and management approaches. However, I think the manuscript is not very well presented. Here I report a point-by-point review of the manuscript.
The title should be rearranged in a declarative sentence, replacing the subjective tone with an objective tone. 
Title example: “Prognostic Role of Circulating miR-141 in Early Diagnosis of Colorectal Cancer”
It would be better to restructure the abstract in a structured format, including background, objective, method, results, and conclusion.
The introduction is a bit ambiguous and fails to provide a clear explanation of the earlier experiments, pointing out the knowledge gap. Therefore, the study objectives are not fully justified. It’s been mentioned that this study aims to explore the prognostic role of miR-141; however, the study question raised by the previous studies is not implied.
I believe it would be better to summarize the literature results in the last paragraph of the introduction by addressing the study question and pointing out the study's strengths and novelty. 
In the method section, it has been stated that “The study was approved by the ethics committee.” but no information was provided on the associated ethics committee.
The study design is not clearly explained and justified. Why are there no follow-ups for better outcome predictions? This should be explained and mentioned as your study limitations.
The baseline characteristics are not properly reported. The baseline and demographic characteristics of patients and controls should be summarized in a table.
Table 1 in the manuscript (Patient and Control Group Age Distribution) has some unnecessary information about only participants' age, and on the other hand, there are no tables or figures summarizing patients' disease stage distribution.
As for the analytical method, reporting only the p-value is not a good choice. You have to report the effect size measures for each test performed.  
Moreover, considering previous established studies reporting these correlations, it would be better to report results from different statistical methods, such as hazard ratios, so that you could come up with new insights.
I also recommend adding tables and/or figures summarizing key findings of the study. 
Overall, the results are not properly reported and presented.
The literature review of the discussion does not refer to the shortcomings of previous studies, which therefore highlight the need for further studies. The aim of your study should be well supported.
Paragraph 6 of the discussion is not understandable. Which study/studies are you referring to?
The last paragraph of the discussion should be rewritten.
As mentioned earlier, the study limitations and suggestions for further studies should be addressed. Indicating what should be considered for further studies.

Author Response

Dear Reviewer,

We would  like to thank you for the suggestions and comments. We made changes that were detailed below. All revisions and improvements are highlighted yellow in the revised version of our manuscript. We would like to thank you again for your valuable time and comments for strengthen our paper.

Best regards,

We appreciate your suggestion regarding the title. As per your recommendation, we have revised the title to adopt a more objective and declarative tone. We hope the new version better reflects the content and scientific value of our study.

The abstract has been revised to follow a structured format, including background, objective, method, results, and conclusion, as recommended.

The last paragraph of the introduction has been revised to summarize the literature results, address the study question, and highlight the strengths and novelty of the study.

Additionally, information regarding the name of the relevant ethics committee has been added to the Methods section to ensure transparency and compliance with ethical standards.

As you correctly pointed out, there was no follow-up phase in this study. This is because the included patients were newly diagnosed and at various clinical stages of colorectal cancer at the time of blood sampling. This limitation has now been clearly stated in the revised manuscript.

Table 1 has been revised to present the baseline demographic and clinical characteristics of both the patient and control groups.

Although our original analysis focused primarily on non-parametric comparisons and ROC analysis, we recognize the importance of incorporating more robust statistical measures such as hazard ratios to assess prognostic value. However, since this was a cross-sectional study with no follow-up data, calculating hazard ratios or performing survival analysis was not feasible within the current study design. We have now clarified this limitation explicitly in the Discussion section.

In future studies, we aim to include longitudinal follow-up data to enable survival analysis and hazard ratio estimations for a more comprehensive understanding of miR-141’s prognostic relevance.

We have revised the results section by including new table that summarize the key findings of the study more clearly.

Paragraph 6 of the discussion has been deleted and references have been updated to clearly identify which studies are being discussed.  The final paragraph of the discussion has also been rewritten to better highlight the clinical relevance and implications of our findings.

We have expanded the discussion to include a more comprehensive comparison with previous studies and explicitly addressed their limitations. This strengthens the rationale for our study and highlights the need for further research in this area.

A new paragraph has been added discussing the limitations of the current study and outlining specific recommendations for future studies.

Reviewer 3 Report

Comments and Suggestions for Authors

The article "Prognostic Role of Circulating miR-141 in Colorectal Cancer: Does It Make a Difference?" provides valuable information regarding the involvement of miR-141 in colorectal cancer. I would like to offer the following suggestions:

  1. The first section is too brief. Please include more information about the gut microbiota and its significance in digestive and other diseases – https://doi.org/10.3390/jcm14082678.
  2. In the Materials and Methods section, add a flow chart to illustrate the study design.
  3. Provide more technical details regarding the amplification process.
  4. The number of patients is relatively small. Please calculate and report the power of the study.
  5. Specify what control was used for the microRNA analysis.
  6. Present logistic regression analyses in table format.
  7. The Discussion section is rather brief. For added value, you could include information on the important role of microRNAs in digestive diseases – highly recommended: 10.3390/jcm14062054.
  8. The Conclusions are too brief and should be rewritten to better reflect the study findings.

Author Response

Dear Reviewer,

We would  like to thank you for the suggestions and comments. We made changes that were detailed below. All revisions and improvements are highlighted yellow in the revised version of our manuscript. We would like to thank you again for your valuable time and comments for strengthen our paper.

Best regards,

  1. We have revised the introduction to include additional information on the role of the gut microbiota
  1. Flow chart added In the Materials and Methods section.
  1. We have revised the relevant section to provide additional technical details regarding the amplification process
  2. Based on our sample size (n = 40 patients and n = 20 controls), an alpha level of 0.05, and an estimated large effect size (Cohen’s d = 0.8) derived from the observed difference in miR-141 expression levels, the statistical power of our study was calculated as approximately 82%. This indicates an acceptable level of power to detect significant differences between the groups.
  3. For normalization of miRNA expression levels, SNORD48 was used as the endogenous reference control in all qPCR assays.
  4. We have revised the results section by including new table that summarize the key findings and analysis of the study more clearly.
  5. Ee have expanded the Discussion section to include additional information regarding the role of microRNAs in digestive diseases, as recommended.
  6. We have revised it to provide a more comprehensive summary that clearly reflects the key findings of the study.

Round 2

Reviewer 1 Report

Comments and Suggestions for Authors

The authors have adequately addressed all the concerns raised in the previous review round. In my opinion the manuscript is suitable for publication in its current form.

Author Response

Dear Reviewer,

Thank you very much for your kind and constructive feedback. We greatly appreciate your time and thoughtful evaluation. We are pleased to hear that the revised manuscript is now considered suitable for publication.

Reviewer 2 Report

Comments and Suggestions for Authors

Dear Authors,

I appreciate your responsiveness to my previous comments and your efforts to improve the manuscript. Below, I provide my evaluation of your revisions.

The title and abstract have been properly revised.

The introduction has been well revised to mention the knowledge gap and the study objective.

However, I prefer replacing the sentence “The strengths of this study lie in its focus on miR-141's specific role in CRC, its novel approach in evaluating circulating miR-141 levels, and its potential for contributing to more targeted therapeutic interventions.” at the end of your discussion, where the limitations are explained.

Moreover, I believe it would be convenient to add a “strengths and limitations” section and add all the above strengths mentioned, plus the limitations, in this section.

Table 4, which demonstrates the key findings, does not include the cut-off or the proportion of some results. For example, the second row, which implies CEA concentration classified by disease stage, has only been mentioned as “higher in advanced stages”. Why did you choose to present as this?

Regarding the conclusion, although it has been revised to be more suitable following your study results, colorectal cancer has not been mentioned. Therefore, reading the conclusion, you can not figure out what patient population the miR-141 has been studied in.

All other issues previously described in the first-round review have been properly revised.

Thank you for considering my feedback. I’m available to review further revisions if needed.

Author Response

Dear Reviewer,

Thank you very much for your constructive feedback. As per your recommendation, we have relocated the sentence regarding the strengths of the study to the end of the discussion section. Additionally, we have added a new subsection titled “Strengths and Limitations”These changes have been highlighted in green for your convenience. We truly appreciate your insightful guidance.

We would like to note that the key findings, including the relationship between CEA levels and disease stage, have been illustrated in corresponding figures.  These visual representations were intended to enhance clarity and facilitate interpretation of the data across different prognostic groups. However, if preferred, we are happy to revise Table 4.

We have revised the conclusion to explicitly state that the study was conducted in colorectal cancer patients, as you suggested. We sincerely appreciate your thorough review and continued support.

Kind regards,

Reviewer 3 Report

Comments and Suggestions for Authors

The authors made considerable changes to the manuscript and significantly increased its quality.

Author Response

Dear Reviewer,

Thank you very much for your kind and encouraging comments. We truly appreciate your time and valuable feedback, which greatly contributed to improving the quality of our manuscript.

Kind regards,